# OPENTAB: ADVANCING LARGE LANGUAGE MODELS AS OPEN-DOMAIN TABLE REASONERS

**Kezhi Kong**[1*]  **Jiani Zhang**[2†]  **Zhengyuan Shen**[2]  **Balasubramaniam Srinivasan**[2]
**Chuan Lei**[2]  **Christos Faloutsos**[2]  **Huzefa Rangwala**[2‡]  **George Karypis**[2]
[1]University of Maryland, College Park    [2]Amazon Web Services
kong@cs.umd.edu  {zhajiani,donshen,srbalasu}@amazon.com
{chuanlei,faloutso,rhuzefa,gkarypis}@amazon.com

## ABSTRACT

Large Language Models (LLMs) trained on large volumes of data excel at various natural language tasks, but they cannot handle tasks requiring knowledge that has not been trained on previously. One solution is to use a retriever that fetches relevant information to expand LLM's knowledge scope. However, existing textual-oriented retrieval-based LLMs are not ideal on structured table data due to diversified data modalities and large table sizes. In this work, we propose OPENTAB, an open-domain table reasoning framework powered by LLMs. Overall, OPENTAB leverages table retriever to fetch relevant tables and then generates SQL programs to parse the retrieved tables efficiently. Utilizing the intermediate data derived from the SQL executions, it conducts grounded inference to produce accurate response. Extensive experimental evaluation shows that OPENTAB significantly outperforms baselines in both open- and closed-domain settings, achieving up to 21.5% higher accuracy. We further run ablation studies to validate the efficacy of our proposed designs of the system.

## 1 INTRODUCTION

The field of natural language processing has rapidly advanced in tasks of text generation (Brown et al., 2020) and knowledge reasoning (Huang & Chang, 2023), driven by general-purpose large-scale generative models like GPT4 (OpenAI, 2023). However, these generative models have inherent drawbacks, like hallucinations (Ye & Durrett, 2022) and limited specialized knowledge. They cannot be used to perform tasks involving data that has not been trained on, like answering questions about recent events or about proprietary enterprise data. Recent advancements in retrieval-based methods (Asai et al., 2023) that aim to equip generative models with expanded information, allows for grounded responses based on real-time or proprietary data.

Many retrieval-based LLMs primarily target textual data from the web or text corpora, overlooking the wealth of information stored in structured tables. When applied to the tables, retrieval-based LLMs encounter the following challenges: (1) Structured tables have diverse data types, notably numerical data presented in large or precise numbers, which results in a high token usage and potentially leads to memory and computational constraints for the model. (2) LLMs, mainly those optimized for natural language understanding, fail to comprehend the complex relationships within tabular data for effective data transformation and answer extraction. (3) The restricted maximum context length of LLMs poses difficulties in managing a multitude of retrieved tables of varying sizes, particularly when dealing with tables containing millions of rows.

This work aims to develop an LLM-based framework for the open-domain table reasoning task (Herzig et al., 2021; Kweon et al., 2023). Here, we concentrate on two specific tasks: table-based question answering (QA) and fact verification. For both tasks, the framework is presented

---

*Work conducted during an internship at Amazon

†Corresponding author

‡Huzefa Rangwala is on LOA as a Professor of Computer Science at George Mason University. This paper describes work performed at Amazon.

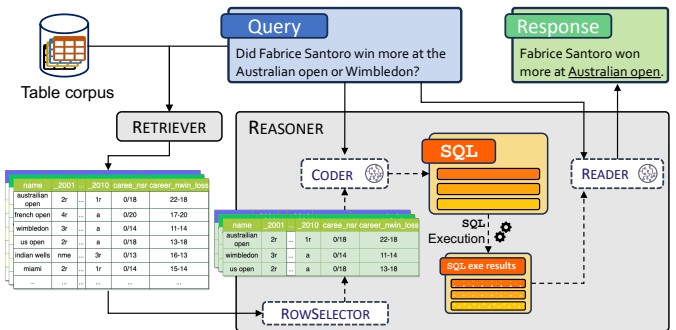 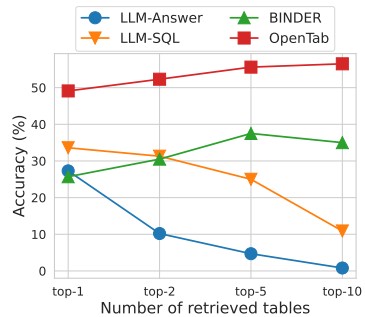

Figure 1: An overview of OPENTAB pipeline. OPENTAB uses a RETRIEVER to retrieve relevant sampled tables from a corpus of tables for a given natural language query, and then use a REASONER to output a natural language response.

Figure 2: The accuracy with an increasing numbers of retrieved tables. OPENTAB has a consistent increase in accuracy with more tables.

with natural language queries and a corpus of tables. In the open-domain setting, the gold tables, which serve as the evidence for queries to generate responses, are not provided. Therefore, the challenge lies in automatically identifying and retrieving pertinent knowledge from the table corpus and also formulating a coherent natural language response. A task example of table QA is in Figure 1.

Existing LLM-based table reasoning methods, like BINDER (Cheng et al., 2023) and DATER (Ye et al., 2023), operate in the closed-domain setting, in which table retrieval is not required and a pre-defined gold table is available for each query during test time. While this assumption simplifies the reasoning process, it does not hold in most real-world scenarios as manually providing gold grounding information is costly and impractical. In the open-domain context, predictions must be grounded in a selection of relevant tables, and the system should possess the reasoning capabilities to sift through incorrect tables (Liu et al., 2023b) to derive the correct output. Meanwhile, existing open-domain table reasoning systems (Chen et al., 2021b; Karpukhin et al., 2020; Chen et al., 2022) have a fine-tuned encoder for retrieval and a fine-tuned table reasoner (Herzig et al., 2020) for QA as well, which necessitates specialized training for specific tasks and is less flexible. Moreover, although they perform well on observed data after fine-tuning, the performance drops on new tables due to constrained transferability.

We propose OPENTAB, an open-domain & end-to-end table reasoning framework. As illustrated in Figure 1, OPENTAB leverages a RETRIEVER to fetch relevant tables, generates programs from a CODER as intermediary reasoning steps, and delegates the final solution to a READER. Following thorough empirical assessment, we opt to implement the BM25 (Robertson et al., 2009a) algorithm as the table RETRIEVER because of its scalability and effectiveness. To mitigate challenges of LLMs in processing and understanding structured tables, we utilize an LLM as a CODER to generate high-quality SQL programs for efficient table parsing. Moreover, an LLM-based READER works to formulate the final response based on the SQL execution results. By breaking down the complex reasoning job into programmatic steps, we achieve enhanced accuracy and reliability in open-domain table reasoning. Notably, the CODER and the READER modules are guided by few-shot prompting, without relying on training or fine-tuning. Our approach has several key advantages:

**High Accuracy.** Open-domain table reasoning presents a challenge due to the trade-off between retrieval recall and precision. Higher recall/coverage requires retrieving more tables, but this might introduce noise from less pertinent tables and therefore reduces precision and prediction accuracy. Specifically, we propose a reranking strategy called **Generative Reranking & Sequential Reasoning** to prioritize the tables with higher similarity between the natural language query and the corresponding generated SQL programs to address the trade-off, which enhances prediction accuracy as shown in Figure 2.

**Scalability.** Our approach can efficiently handle large quantities and sizes of tables. The framework is capable of generating accurate responses with access to only a limited number of rows from each table effectively extracted by ROWSELECTOR.

**Robustness**. We propose *simple-to-complex* prompting, a flexible and robust progressive program generation and execution strategy. By sequentially generating SQL programs with increasing levels of complexity, starting from basic column selection to advanced operations like aggregation and

text operations, the model explores a wider range of possible solutions. This approach offers more adaptability in finding the optimal SQL query for a given task, and it also serves as a structured reasoning process for generating effective SQL code. Additionally, it allows for easy fallback to simpler queries if the more complex ones fail, enhancing the overall robustness of the system.

**Effectiveness.** We provide extensive experimental results showing that our method registers competitive performances without any fine-tuning on the target dataset. We show that OPENTAB significantly outperforms baselines in both open- and close-domain settings. For example OPENTAB outperforms the best baseline with a remarkable 21.5% higher accuracy under the open-domain scenario. We further run detailed ablation studies to validate the efficacy of our proposed designs.

## 2 DEFINITION OF OPEN-DOMAIN TABLE REASONING

We formally define the open-domain table reasoning task as follows: Given a natural language input query $q$ and a database with tables $\mathcal{T}$, the task is to generate a response $a$ based on multiple retrieved tables, denoting as $\widehat{\mathcal{T}_q} \subset \mathcal{T}$, while the gold table set for query $q$ is denoted as $\mathcal{T}_q$. The form of the $(q, a)$ pair is determined by the corresponding downstream task. In this work, we focus on the following two tasks:

- **Question answering**, where $q$ is a natural language question and $a$ is a natural language answer. E.g., $q$: "Which team won the NBA championship in 2022?", $a$: "Golden State Warriors."
- **Fact verification**, where $q$ is a statement and $a$ is a judgement given $\mathcal{T}$, like entailment or contradiction. E.g., $q$: "In àlex corretja career there be no game in may 1998", $a$: "Entailed".

We approach the task as modeling $P(a|q, \mathcal{T})$. Technically, as $\mathcal{T}$ can be a large collection of tables, it is intractable to directly solve $P(a|q, \mathcal{T})$. Thus, we follow the two-step approach that is often adopted in natural language retrieval augmented models (Borgeaud et al., 2022). Namely, given a query $q$, we firstly retrieve a set of evidence tables $\widehat{\mathcal{T}_q} \subset \mathcal{T}$, and subsequently model $P(a|q, \widehat{\mathcal{T}_q})$ in place of $P(a|q, \mathcal{T})$. In this work, we expect to leverage LLMs as the backbone for modeling given their capability in few-shot inference and reasoning (Chen, 2023).

Note that the key difference between open-domain and closed-domain table reasoning task is that in the closed-domain setting the tables $\mathcal{T}_q$ are explicitly provided for each query $q$. In contrast, in the open-domain setting, there is possibility that $\mathcal{T}_q \cap \widehat{\mathcal{T}_q} = \emptyset$ depending on the ability of the retriever.

## 3 METHOD

**Overview.** We present an overview of OPENTAB in Figure 1. To enable our pipeline to automatically handle large-scale tabular data in terms of both table size (number of rows and columns) and table quantity, we divide the overall system into two components: a RETRIEVER (in Section 3.1) and a REASONER (in Section 3.2). For RETRIEVER, we advocate the BM25 for table retrieval tasks as it provides scalable and competitive retrieval performance. For the REASONER, we leverage the CODER powered by LLM to generate SQL queries based on retrieved tables $\mathcal{T}_q$. The final response $a$ is then extracted by an LLM-based READER module to ensure the accuracy, efficiency, and robustness of the REASONER against generation stochasticity. In Section 3.3, we introduce a Generative Reranking & Sequential Reasoning (GRSR) strategy for open-domain reasoning, which sequentially generates SQLs for a set of retrieved tables and then reranks them based on query similarity, effectively addressing the hallucination issues.

### 3.1 TABLE RETRIEVER

Effective table retrieval in the open-domain setting (Wang et al., 2022; Kweon et al., 2023) remains an open question (Ren et al., 2022). We select BM25 [1] (Robertson et al., 2009b) as the table retriever in our framework for the following considerations. First, BM25, a probabilistic-based ranking function used in information retrieval systems to rank documents, is a standard sparse retrieval method that works efficiently on large corpora, considering term frequency and inverse document frequency

---

[1] Implementation based on https://github.com/dorianbrown/rank_bm25

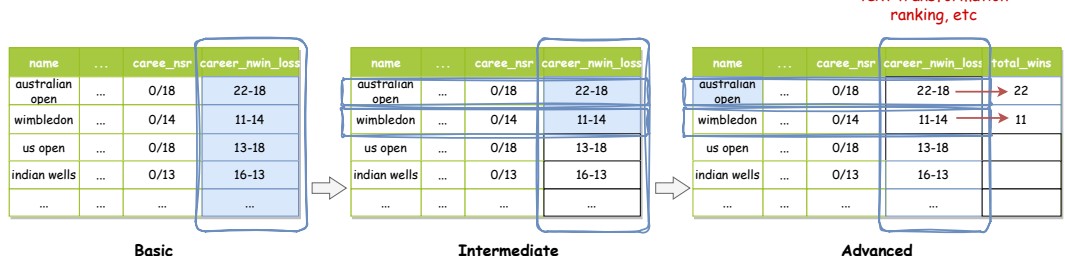

Figure 3: Examples illustrating the progressive *Simple-to-complex* SQL proficiency. The *basic* SQL queries mainly select specific columns. The *intermediate* incorporates both column and row selection. The *advanced* SQL employs additional operations like aggregation and text operations that can manipulate and transform the tabular data. The cells in blue are outputs of the SQL programs.

with saturation. Second, it is easy to use without the necessary fine-tuning process of Dense Passage Retrieval (DPR) (Karpukhin et al., 2020). Third, empirically, we show that BM25 achieves competitive retrieval performance compared with dense methods such as DPR in Section 4.3. Given these, BM25 as RETRIEVER together with REASONER powered by LLMs make the OPENTAB system off-the-shelf usable without the necessity of fine-tuning on target databases, which is a significant advantage for industrial applications.

## 3.2 TABLE REASONER

The REASONER processes natural language queries along with table schema and sampled rows, facilitated by a CODER and a ROWSELECTOR. The ROWSELECTOR ensures that the pertinent rows are provided to the LLM for effective processing. The CODER generates SQL programs of increasing complexity. Finally, the READER leverages an LLM to parse and extract the final response.

### 3.2.1 CODER

On the table reasoning task, LLMs (e.g., ChatGPT and Falcon) are capable of generating SQL queries that can be executed on databases to extract the final answer (Chen et al., 2021a; Liu et al., 2023a; Sun et al., 2023). This approach is referred to as the LLM-SQL baseline. However, the generated SQLs towards direct question answering can be less robust with both syntax and semantic errors during execution when facing complex reasoning tasks (Cheng et al., 2023), which might require multiple operations such as aggregation, comparison, and sorting (Kweon et al., 2023).

In CODER, we propose a new ***simple-to-complex*** prompting strategy for effective SQL generation. Specifically, for each input query, we prompt the LLM to sequentially generate three SQL programs with ascending complexities and functionalities, which are:

- *SQL-basic*: focusing on column selection, which sets the groundwork for understanding how to fetch specific data from a database.

- *SQL-intermediate*: incorporating both column and row selection. This means extracting particular columns and filtering rows based on specified criteria, enhancing precision in data gathering.

- *SQL-advanced*: empowering additional operations including but not limited to aggregation functions and text operations. Aggregation empowers data summarization, while text operations facilitate the manipulation and transformation of string data.

Figure 3 shows the high-level idea of this *simple-to-complex* prompting strategy. The reason for such a design is that generating solid SQLs can be challenging for complex input queries, so we incrementally generate SQLs in reasoning steps from simple to complex. Moreover, with the READER module (explained later), the generated SQL programs do not necessarily need to produce the final answer directly. This is infeasible when the input query is not solvable by pure SQL programs (Cheng et al., 2023). Therefore, *SQL-basic* and *SQL-intermediate* can not only function as reasoning steps towards the ultimate solid *SQL-advanced* but can also lead to a final correct response.

This provides more flexibility and robustness than generating a single SQL. After the generations of three SQL programs, we first test *SQL-advanced* by execution as verification. We proceed to the READER module for final response extraction if it returns valid non-empty results. If the SQL fails the verification, we turn to verify simpler ones until all SQLs are exhausted. Figure 5 displays the prompt and generation structure of CODER. For concrete problem solving example, see Figure 6.

### 3.2.2 READER

To expand the limited capability of SQL in solving natural input queries, we use a READER module that leverages LLM to digest the intermediate SQL execution results and formulate the final response. Rather than providing only the execution results, we further supply the READER with broader context from CODER, including table schema, sampled rows, and generated SQL query. This enables the READER to better understand the contextual background and semantics needed for accurate predictions. We opt to provide READER with these additional contexts based on analysis showing poor performance when only giving the execution results without exposure to the richer contextual information from CODER (see ablation studies). As shown in Figure 5, our framework prompts the READER with this broader context encapsulating the entire SQL generation process.

### 3.2.3 ROWSELECTOR

Besides table schema, table contents are also critical for effective SQL generation. Ideally, the entire table should be input to the LLM for the complete view. However, LLMs can only handle tables with large sizes within their token capacity limits. In order to balance scalability and reasoning ability, we propose to use the ROWSELECTOR to harness a few rows that are most relevant to the input $q$ to be placed into the prompt of LLMs. Technically, we leverage BM25 to rerank and choose the top-$k$ rows based on relevance to $q$.

### 3.3 OPEN-DOMAIN TABLE REASONING

In contrast to a closed-domain setting where the table used for reasoning is provided, open-domain reasoning operates over a much larger and unconstrained table store, causing an inherent trade-off between precision and recall of the retrieved tables. More tables need to be retrieved to achieve higher coverage (i.e., recall values), which inevitably brings irrelevant information and, thus, lower precision. Therefore, precisely identifying the correct table from the retrieved set is crucial for the pipeline to achieve accurate reasoning performance over the end-to-end open-domain task.

We propose a novel **Generative Reranking & Sequential Reasoning (GRSR)** strategy to address this pain point. Specifically, given a query $q$ and $\widehat{\mathcal{T}}_q$ of $k$ tables fetched by RETRIEVER, SQLs are sequentially generated for each table using CODER. We then rerank the tables based on the similarity between $q$ and the generated SQL computed using *pretrained* Cross Encoder transformers (Reimers & Gurevych, 2019). [2] The tables with the highest resultant similarity scores computed by corresponding SQLs are then selected for downstream predictions. GRSR is effective because of its ability to combat the hallucination tendency of LLMs. Ideally, CODER should only be able to generate valid SQL programs if gold set $\mathcal{T}_q$ is retrieved, such that the execution result of the SQL can be used to locate the gold tables. However, in practice, we find that CODER is likely to generate valid SQL that can pass the execution verification even though the retrieved table is irrelevant from the query $q$. This hallucination problem greatly limits the expected effectiveness of generative feedback from LLMs toward retrieved tables. Given these, GRSR evades noisy retrieved tables by detecting hallucinated SQLs using a pretrained Cross Encoder. With more tables retrieved, we expect to see higher recall value but also likely with precision degradation. GRSR effectively counters such negativity introduced by more tables retrieved while leveraging the advantage of higher recall by more precisely locating the gold table and ignoring the noisy ones.

To better reveal the efficacy of the GRSR strategy, we introduce two baseline algorithms for comparison. **Joint Reasoning** approach, where all $k$ retrieved tables are aggregated into the prompt of CODER for SQL generation and prediction simultaneously. **Sequential Reasoning** only, which is a basic sequential approach that verifies the validity of SQLs in the order defined by the retriever. If we reach a valid SQL without an execution error, we select the corresponding SQL. By default,

---

[2]Huggingface checkpoint with model name `cross-encoder/ms-marco-MiniLM-L-12-v2`

Table 1: The accuracy on two open-domain table reasoning tasks using two different backbone LLMs. The results are presented at the top-1/2/5/10 retrieved tables.

| Method | gpt-3.5-turbo | | | | falcon-180B | | | |
|---|---|---|---|---|---|---|---|---|
| | Top-1 | Top-2 | Top-5 | Top-10 | Top-1 | Top-2 | Top-5 | Top-10 |
| | *Open-WikiTables* | | | | *Open-WikiTables* | | | |
| LLM-Answer | 0.273 | 0.102 | 0.047 | 0.008 | 0.203 | 0.047 | 0.016 | 0.008 |
| LLM-SQL | 0.336 | 0.313 | 0.250 | 0.109 | 0.195 | 0.094 | 0.023 | 0.008 |
| BINDER | 0.234 | 0.305 | 0.375 | 0.375 | 0.164 | 0.188 | 0.227 | 0.227 |
| OPENTAB | **0.491** | **0.523** | **0.556** | **0.565** | **0.375** | **0.391** | **0.437** | **0.437** |
| | *FEVEROUS* | | | | *FEVEROUS* | | | |
| LLM-Answer | 0.672 | **0.508** | 0.492 | 0.453 | 0.680 | 0.476 | 0.375 | 0.289 |
| LLM-SQL | 0.609 | 0.460 | 0.515 | 0.422 | 0.328 | 0.242 | 0.195 | 0.094 |
| BINDER | 0.375 | 0.383 | 0.391 | 0.391 | 0.219 | 0.266 | 0.305 | 0.281 |
| OPENTAB | **0.695** | **0.508** | **0.563** | **0.508** | **0.742** | **0.515** | **0.555** | **0.484** |

we leverage GRSR in OPENTAB. In the ablation section, we show the performance gain of GRSR over Sequential Reasoning and Joint Reasoning, justifying the efficacy of GRSR in mitigating the hallucination problem and locating the gold table.

## 4 EXPERIMENTS

### 4.1 EXPERIMENT SETUP

**Datasets**. To evaluate the proposed approach, we use **Open-WikiTable** (Kweon et al., 2023), **WikiTableQuestions** (Pasupat & Liang, 2015), and **FEVEROUS** (Aly et al., 2021) datasets. Open-WikiTable is an open-domain table question-answering dataset, where the specific table related to a given question is not provided and must be retrieved. Experiments were limited by budget to 2,000 random samples from the validation set. The corpus contains 24,680 candidate tables. In contrast, WikiTableQuestions, also a table-based QA dataset, provides the relevant table containing the necessary information, making it closed-domain. FEVEROUS, on the other hand, is a fact-verification dataset that requires the identification of multiple relevant tables for reasoning and verifying factual questions. We adapted the dataset for open-domain table reasoning by filtering 323 examples from the validation set that rely solely on table data, using a corpus of 26,177 candidate tables from the FEVEROUS dataset. In line with BINDER (Cheng et al., 2023), we use execution accuracy (EA) as our primary metric for evaluating performance on the Open-WikiTable and WikiTableQuestions datasets. For FEVEROUS we adopt a metric that demands the pipeline not only to make the correct predictions but also to correctly identify the gold evidence table(s) for making those predictions. We report both metrics as the accuracy term in the following descriptions. See details in Appendix A.3.

**Baselines**. For table retrieval experiments, we conduct experiments to compare the performances of BM25 and DPR, where DPR uses BERT models[3] both with and without being fine-tuned on the target dataset. In the context of the table reasoning task, we compare with BINDER (Cheng et al., 2023), an end-to-end table QA model that generates specialized symbolic languages to be executed on the table database. As the original LLM model used in BINDER (OpenAI Codex) is no longer available, we reproduce BINDER's results using gpt-3.5-turbo with BINDER's official implementation. The official implementation of BINDER does not support the open-domain scenario; and the database interface cannot support identification of multiple relevant tables. We leverage the straightforward Sequential Reasoning (SR) strategy to reduce the multi-table scenario to single-table to make BINDER applicable in the open-domain setting. Furthermore, we implement two additional baselines: (1) LLM-Answer, to prompt LLM to directly output the text-format answer based on the table and question; (2) LLM-SQL, to let LLM respond with SQL programs whose execution result will be the output. If not specified, the LLM backbone used in this work is gpt-3.5-turbo (default 4k-token version), and the in-context learning examples are 2-shot.

---

[3]Huggingface checkpoint with model name bert-base-uncased

Table 2: The accuracy on the closed-domain table QA task (WikiTableQuestions).

| | Method | Acc |
|---|---|---|
| Fine-tuned | T5-3B (Xie et al., 2022) | 0.519 |
| | Tapex (Liu et al., 2022) | 0.604 |
| | TaCube (Zhou et al., 2022) | 0.611 |
| | OmniTab (Jiang et al., 2022) | 0.633 |
| w/o Fine-tuning | LLM-Ans | 0.375 |
| | LLM-SQL | 0.414 |
| | BINDER (Cheng et al., 2023) | 0.428 |
| | DATER (Ye et al., 2023) | 0.453 |
| | OPENTAB | **0.641** |

Table 3: The accuracy results on the closed-domain table QA task (Open-WikiTable).

| Method | Acc |
|---|---|
| LLM-Answer | 0.378 |
| LLM-SQL | 0.579 |
| BINDER (Cheng et al., 2023) | 0.421 |
| OPENTAB | **0.710** |

## 4.2 MAIN RESULTS

**Open Domain**. For this setting, we first retrieve top-$k$ relevant tables $\widehat{\mathcal{T}}_q$ from the database using the same BM25 algorithm. We run extensive experiments with 2 different LLM backbones (`gpt-3.5-turbo` & `falcon-180b`) on two datasets (Open-WikiTables & FEVEROUS). Table 1 summarizes the experimental results. From Table 1, we can see that OPENTAB significantly outperforms baselines on the Open-WikiTables dataset. For example with `gpt-3.5-turbo` OPENTAB outperforms the best baseline with a remarkable 21.5% higher accuracy under the top-10 scenario. Meanwhile on the FEVEROUS dataset, OPENTAB can mostly win over the baselines with non-trivial improvements. Note that the baseline methods require full tables to be fed into LLM for reasoning, which may exceed the upper bound of the input token limit, rendering an invalid prediction. While OPENTAB does not suffer from scalability issues as CODER is capable of generating high-quality SQLs with three selected representative rows. Moreover, to reproduce BINDER's performance, we follow Cheng et al. (2023) to use 14-shot examples when doing in-context learning. For BINDER, we also deploy `gpt-3.5-turbo-16k` to fit the long input sequences. This group of experiments shows that OPENTAB is the more capable system at dealing with the challenging open-domain table reasoning tasks.

**Closed Domain**. To complement the empirical investigation, we further conduct experiments under the closed-domain scenario, where the golden evidence table is directly given without the retrieval requirements. We show the performances in Table 3 on the Open-WikiQuestion dataset. Similarly, to make BINDER applicable, we use `gpt-3.5-turbo-16k` with its official 14-shot examples. We can see that OPENTAB has a non-marginal 13% improvements over the baselines, further validating the efficacy of OPENTAB designs. We further evaluated the WikiTableQuestions dataset (Pasupat & Liang, 2015), which is intrinsically a closed-domain table QA dataset. Results are shown in Table 2. It reveals the competitiveness of OPENTAB of even outperforming fine-tuned methods.

## 4.3 TABLE RETRIEVAL RESULTS

We report the table retrieval performance in Table 4. Results on the Open-WikiTable dataset are from both Kweon et al. (2023) and our implementations. BM25 has competitive performances compared to the DPR, which has been fine-tuned on the labeled training dataset. On the FEVEROUS dataset, BM25 even outperforms fine-tuned BERT, which supports our intuition that sparse retrievers are simple but powerful table retrievers.

## 5 ABLATION STUDIES & ANALYSIS

**Ablation studies on the proposed modules of OPENTAB.** In Table 5, we show the effects of designed components our OPENTAB. The evaluations happen under the closed-domain scenario. We can see that, the performance drops drastically from 0.71 to 0.54 due to the removal of the STC module, highlighting its significance. Moreover with the ablation of ROWSELECTOR, the performance continues to drop. Lastly, by disabling the longer-context functionality of READER, the performance reaches the lowest, showing the significance of incorporating detailed context information including

Table 4: Recall@$k$ scores for table retrieval on the open-domain Open-WikiTables and FEVEROUS datasets. BM25 wins among the non fine-tuned methods.

| Setting | Method | Recall@5 | Recall@10 | Recall@20 | Recall@50 |
|---|---|---|---|---|---|
| **Open-WikiTables** | | | | | |
| w/o Fine-tuning | BM25 | **0.832** | **0.893** | **0.940** | **0.972** |
| | BM25* | 0.422 | 0.489 | 0.561 | - |
| | DPR-BERT | 0 | 0 | 0 | 0 |
| Fine-tuned | DPR-BERT* | **0.895** | **0.950** | 0.973 | **-** |
| | DPR-BERT | 0.870 | 0.933 | **0.975** | 0.990 |
| **FEVEROUS** | | | | | |
| w/o Fine-tuning | BM25 | **0.919** | **0.942** | **0.956** | **0.973** |
| | DPR-BERT | 0 | 0 | 0.001 | 0.003 |
| Fine-tuned | DPR-BERT | 0.698 | 0.767 | 0.831 | 0.902 |

[1] Results of *methods are from Kweon et al. (2023). Others are implemented in this work.
[2] The non fine-tuned DPR scores are approximate to 0 on both datasets due to poor transferability.

Table 5: Ablation studies on different proposed modules in OPENTAB. On the closed-domain Open-WikiTables dataset. STC stands for the *Simple-to-Complex* prompting SQL generation strategy.

| Method | Accurate |
|---|---|
| OPENTAB | **0.710** |
| OPENTAB w/o STC | 0.539 |
| OPENTAB w/o STC+ROWSELECTOR | 0.515 |
| OPENTAB w/o STC+ROWSELECTOR +*broad-context* READER | 0.421 |

that of the SQL generation process. For clarification, by disabling the longer-context functionality, we do not forward the table schemas and generated SQLs into READER.

**Ablation studies on *Simple-to-Complex* SQLs.** We test the individual efficacy of *SQL-basic*, *SQL-intermediate*, and *SQL-advanced*. Instead of moving onto the next in order if one program fails to provide valid result, we focus on one SQL program at a time for evaluation individually. Table 6 show the results. We can see that our design of synergizing all three kinds of design has the best performance compared with merely using solo SQLs. This is because in our *Simple-to-Complex* strategy, the solidness of the generated SQL programs will be verified on the fly, thus we can try to evade invalid SQLs that will either have syntax errors or empty execution results, which is common for a solo SQL generation.

**Open-domain strategy.** We verify the efficacy of the Joint Reasoning, Sequential Reasoning, and GRSR strategies. Results are summarized in Figure 4. From the plot we can see that the best performing model is OPENTAB with GRSR strategy, which is intuitive because such strategy implicitly leverages the generative power of the LLM towards identifying the correct table to build reasoning on. From the figure we can see that the accuracy of OPENTAB-GRSR constantly increases with more tables retrieved (higher recall value) despite the negative impact of more incorrect tables presented. The novel generative reranking design leads to more precisely grabbing the core information by mitigating the hallucination issue.

## 6   RELATED WORKS

**LLMs for Table Reasoning.** Recent work has explored leveraging LLMs for table reasoning tasks without task-specific fine-tuning. Chen (2023) showed that LLMs like GPT-3 can perform substantially on table QA and fact verification datasets like WikiTableQuestions and TabFact with few-shot prompting. However, their capability degrades on large tables with 30+ rows. To address this limitation, Ye et al. (2023) proposed using LLMs to extract relevant sub-tables and decompose complex questions into simpler sub-questions. Their DATER method improves LLM performance on large

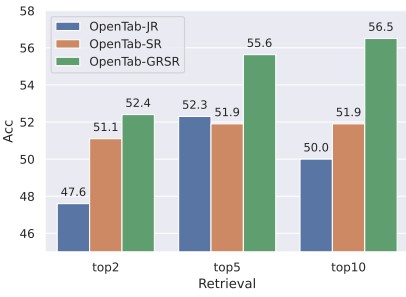

Figure 4: Ablation study on the open-domain strategy. "JR" stands for "Joint Reasoning" and "SR" stands for "Sequential Reasoning".

Table 6: Ablation studies on the STC SQL generations.

|  | Closed-domain | Open-domain-top2 |
|---|---|---|
| OPENTAB | **0.710** | **0.523** |
| OPENTAB-adv. | 0.585 | 0.367 |
| OPENTAB-int. | 0.609 | 0.406 |
| OPENTAB-basic | 0.617 | 0.390 |

and complex table reasoning tasks. Alternatively, BINDER (Cheng et al., 2023) maps the natural language questions into programing languages such as SQL and Python with API calls to invoke LM functions, improving the reasoning capability beyond the basic programming language grammar. He et al. (2022) proposed a strategy that involves rethinking with retrieval to enhance LLM faithfulness by retrieving relevant knowledge using the chain-of-thought prompting, showing the effectiveness in a series of reasoning tasks, including tabular reasoning.

**Table Retrieval.** Wang et al. (2022) investigated the necessity of designing table-specific dense retrievers with structure-related modules. The authors showed that DPR (Dense Passage Retriever) and DTR (Dense Table Retriever) have comparable performances, justifying the effectiveness of deploying DPR in table retrieval tasks. Furthermore, Kweon et al. (2023) shared similar findings on the Open-WikiTables dataset. Despite the efforts of applying dense retrievers on tables, a number of existing works rely on sparse methods to realize the retriever. Schlichtkrull et al. (2021) leveraged TF-IDF to retrieve tables in the open-domain setting. Aly et al. (2021) introduce the FEVEROUS dataset and also use a sparse method DrQA (Chen et al., 2017) to retrieve both text and tables. Moreover, DCUF (Hu et al., 2022) utilized DrQA to retrieve and BM25 to rerank the retrieved documents. Chen et al. (2022) proposed to use fine-tuned DPR on open-domain table retrieval task, and the answers are heavily dependent on the generated SQL queries. These dependencies can result in subpar retrieval performance for unseen tables and cannot handle the cases when the generated SQL queries fail to execute. In summary, effectively retrieving relevant tables from a large data corpus still remains an open question.

**Retrieval Augmented Generators (RAG).** RAGs leverage retrievers to fetch information from external knowledge database and augment the text input to the language models (Lewis et al., 2020). REALM (Guu et al., 2020) and RETRO (Borgeaud et al., 2022) pretrains the retrieval augmentation framework in the end-to-end manner so that the language model learns to better utilize the retrieved information, while ATLAS (Izacard et al., 2022) jointly trains the retriever as well as the language model. We point out that these well-known RAG models are specialized in the text reasoning domain and cannot be directly applied to table reasoning tasks without fine-tuning, thus inapplicable to the experimental setup of this work.

## 7 CONCLUSION

In this work, we study the task of open-domain table reasoning, where complex questions are answered using knowledge and evidence stored in structured tables. The reasoning part is empowered by Large Language Models (LLMs), enabling the OPENTAB to work directly on the target dataset without being specifically fine-tuned. This is a highly desired property for several applications, from both efficiency and privacy perspectives. The proposed method OPENTAB incorporates CODER, ROWSELECTOR, and READER that are effective in generating valid SQL programs, highlighting informative content and absorbing longer context allowing for accurate response. We perform comprehensive empirical studies demonstrating the efficacy of OPENTAB. In both open-domain and closed-domain settings, OPENTAB outperforms baseline methods by a large margin, revealing its advanced capability.

## 8 REPRODUCIBILITY

To ensure reproducibility of OPENTAB, we provide detailed discussion regarding the experimental setup in Section 4, including the LLM backbone usage, pretrained checkpoint model name, few-shot sample count, etc. We share the prompt used in the experiments in the Appendix. We discuss the details of datasets and the specific metrics used in Section A.3.

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

# A APPENDIX

## A.1 CODER IN-CONTEXT LEARNING PROMPT (2 SHOTS)

```
Given the table schema and three example rows out of the table, write a SQLite program to
    ↪ extract the sub-table that contains the information needed to answer the questions.
The SQLite does not need to directly answer the question.
Assume you always have enough information when executing the SQLite.
If you cannot generate the SQLite with high confidence that it is correct, then generate some
    ↪ SQLite that is less complex but correct.
Try to use fuzzy-match for values if you are not sure about the values.
Generate 3 SQLite programs with respect to the question separated by [SQLSEP], the
    ↪ complexities of the SQLite programs generated ascend (basic, intermediate, advanced).

CREATE TABLE Fabrice_Santoro(
        row_id int,
        name text,
        _2001 text,
        _2002 text,
        _2003 text,
        _2004 text,
        _2005 text,
        _2006 text,
        _2007 text,
        _2008 text,
        _2009 text,
        _2010 text,
        career_nsr text,
        career_nwin_loss text)
/*
3 example rows:
SELECT * FROM Fabrice_Santoro LIMIT 3;
row_id name    _2001  _2002  _2003  _2004  _2005  _2006  _2007  _2008  _2009  _2010
    ↪     career_nsr    career_nwin_loss
0   australian open    2r    1r 3r 2r    1r     qf     3r     2r     3r     1r
    ↪     0 / 18   22  1 8
1    french open    4r     2r     2r     3r     1r     1r     1r     2r     1r
    ↪     a      0 / 20  17  2 0
2     wimbledon     3r     2r     2r     2r     2r     2r     2r     1r     2r
    ↪     a      0 / 14  11  1 4
*/
Q: did he win more at the australian open or indian wells?
SQLite:
SELECT name, career_nwin_loss FROM Fabrice_Santoro; [SQLSEP]
SELECT name, career_nwin_loss FROM Fabrice_Santoro WHERE name LIKE "%australian open%" OR name
    ↪   LIKE "%indian wells%"; [SQLSEP]
WITH Wins AS (
    SELECT
    name,
    CAST(SUBSTR(career_nwin_loss, 1, INSTR(career_nwin_loss, '   ') - 1) AS INT) AS wins,
    CAST(SUBSTR(career_nwin_loss, INSTR(career_nwin_loss, '   ') + 1) AS INT) AS losses
    FROM Fabrice_Santoro
    WHERE name LIKE "%australian open%" OR name LIKE "%indian wells%"
)
SELECT name, SUM(wins) as total_wins, SUM(losses) as total_losses FROM Wins GROUP BY name;

CREATE TABLE Playa_de_Oro_International_Airport(
        row_id int,
        rank text,
        city text,
        passengers text,
        ranking text,
        airline text)
/*
3 example rows:
SELECT * FROM Playa_de_Oro_International_Airport LIMIT 3;
row_id   rank    city    passengers    ranking    airline
0   1    united states, los angeles    14,749    nan    alaska airlines
1   2    united states, houston    5,465    nan    united express
2   3    canada, calgary    3,761    nan    air transat, westjet
*/
Q: how many more passengers flew to los angeles than to saskatoon from manzanillo airport in
    ↪ 2013?
SQLite:
SELECT city, passengers FROM Playa_de_Oro_International_Airport; [SQLSEP]
SELECT city, passengers FROM Playa_de_Oro_International_Airport WHERE city LIKE "%los angeles
    ↪ %" OR city LIKE "%saskatoon%"; [SQLSEP]
```

```
WITH PassengerCounts AS (
    SELECT
    city,
    CAST(REPLACE(passengers, ',', '') AS INT) AS passenger_count
    FROM Playa_de_Oro_International_Airport
    WHERE city LIKE "%los angeles%" OR city LIKE "%saskatoon%"
)
SELECT
SUM(CASE WHEN city LIKE "%los angeles%" THEN passenger_count ELSE 0 END) -
SUM(CASE WHEN city LIKE "%saskatoon%" THEN passenger_count ELSE 0 END) AS passenger_difference
FROM PassengerCounts;
```

## A.2 READER IN-CONTEXT LEARNING PROMPT (2 SHOTS)

```
Given the execution result attained by running SQLite, extract the final answer of the
    ↪ question from the table.
Assume you can always find the answer from the table, so you must give an answer that makes
    ↪ sense to the question based on the given table.
If the answer contains multiple items, separate them by [SEP].

CREATE TABLE Fabrice_Santoro(
        row_id int,
        name text,
        _2001 text,
        _2002 text,
        _2003 text,
        _2004 text,
        _2005 text,
        _2006 text,
        _2007 text,
        _2008 text,
        _2009 text,
        _2010 text,
        career_nsr text,
        career_nwin_loss text)
/*
3 example rows:
SELECT * FROM Fabrice_Santoro LIMIT 3;
row_id  name    _2001   _2002   _2003   _2004   _2005   _2006   _2007   _2008   _2009   _2010
    ↪       career_nsr      career_nwin_loss
0       australian open 2r      1r  3r  2r      1r      qf      3r      2r      3r      1r
    ↪           0 / 18  22  1 8
1       french open     4r      2r      2r      3r      1r      1r      1r      2r      1r
    ↪       a       0 / 20  17  2 0
2       wimbledon       3r      2r      2r      2r      2r      2r      2r      1r      2r
    ↪       a       0 / 14  11  1 4
*/
Q: did he win more at the australian open or indian wells?
SQLite:
SELECT
    name,
    CAST(SUBSTR(career_nwin_loss, 1, INSTR(career_nwin_loss, '-') - 1) AS INT) AS total_wins
FROM
    Fabrice_Santoro
WHERE
    (name LIKE '%australian open%') OR
    (name LIKE '%indian wells%')
Execution Result:
name    total_wins
australian open 22
indian wells    23
A:
indian wells

CREATE TABLE Playa_de_Oro_International_Airport(
        row_id int,
        rank text,
        city text,
        passengers text,
        ranking text,
        airline text)
/*
3 example rows:
SELECT * FROM Playa_de_Oro_International_Airport LIMIT 3;
row_id  rank    city    passengers      ranking airline
0   1   united states, los angeles  14,749  nan alaska airlines
1   2   united states, houston  5,465   nan united express
```

Table 7: Applicability of OPENTAB and baselines.

| Method | Open-domain | Large table | w/o Fine-tuning |
|---|---|---|---|
| FEVEROUS-base (Aly et al., 2021) | ✓ | | |
| DCUF (Hu et al., 2022) | ✓ | | |
| OpenWT-base (Kweon et al., 2023) | ✓ | | |
| BINDER (Cheng et al., 2023) | | | ✓ |
| DATER (Ye et al., 2023) | | | ✓ |
| OPENTAB (proposed) | ✓ | ✓ | ✓ |

```
2    3    canada, calgary    3,761    nan    air transat, westjet
*/
Q: how many more passengers flew to los angeles than to saskatoon from manzanillo airport in
    ↪ 2013?
SQLite:
SELECT
    city,
    REPLACE(passengers, ',', '') AS passenger_count
FROM
    Playa_de_Oro_International_Airport
WHERE
    (city LIKE '%los angeles%') OR
    (city LIKE '%saskatoon%')
Execution Result:
city    passenger_count
united states, los angeles  14749
mexico, saskatoon  10000
A:
4749
```

## A.3 DATASETS

**Open-WikiTables.** We directly leverage the Open-WikiTables dataset provided by (Kweon et al., 2023) in our experiments. Note that this dataset has limited amount of cases whose final ground truth answer is empty, which will not be used in our experiments. Our evaluations happen on the validation set of the dataset. Due to the budget limitation, the experiments are carried out on 2,000 random samples out of the validation set with a fixed random seed 42 to make sure all the models and methods are evaluated on the same subset fairly. Note that for this dataset each input question will only has one gold table as evidence. Table corpus size is 24,680. For the evaluation metric, we adopt the *semantic-match* evaluator as used in BINDER (Cheng et al., 2023), which is a more reasonable metric to evaluate the table QA task.

**FEVEROUS.** FEVEROUS (Aly et al., 2021) dataset is a fact verification dataset consisting of evidences in the format of both natural language text and structured tables. To adapt the dataset into the open-domain table reasoning setting, we filter 323 claims out of the validation set whose verification is only based on table information. Also because the tables are noisy web tables where a lot of samples incorporate undesired properties such as empty sells, multi-header, cell count mismatch, etc. These tables cannot naturally be used in the SQL database interface so we need to further remove claims that need to be verified on such tables. The table corpus of the FEVEROUS dataset used here is of 26,177 different tables. Note that for the processed subset, each claim will only have one gold table as evidence. The prediction space is limited to "refutes" and "supports". For the evaluation metric, inspired by the FEVEROUS score metric in Aly et al. (2021), which requires predictions to be made on the correct evidences, we further invent our metric as to make the right prediction while locating the correct evidence table in the same time. Such scenario will be regarded as a successful prediction. As long as the model does not make the right decision, or makes decision based on the wrong table, it will be considered a failure case.

## A.4 SANITY CHECK ON TEXT-TO-SQL DATASET

STC is proposed in the context of end-to-end table QA task, in which SQL programs work as an efficient and scalable interface to query the database and extract relevant information for READER to make inference on. SQL generation is an intermediate tool in the QA pipeline, instead of outcome. While on text-to-SQL benchmarks, ground truth is SQL program itself, which is different from our

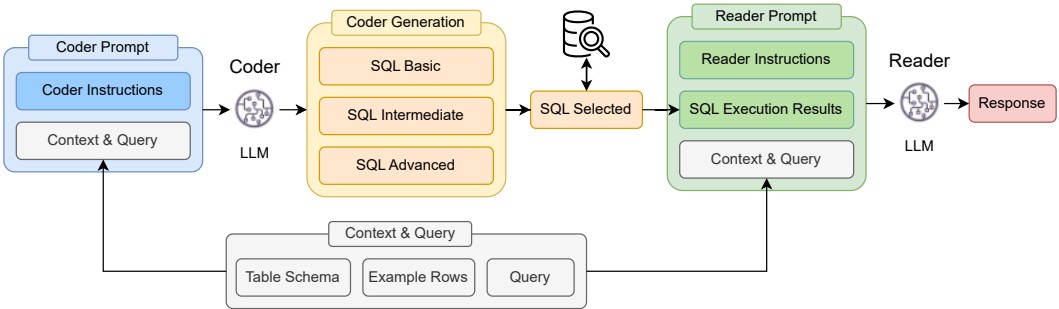

Figure 5: Prompt and generation structures of both CODER and READER.

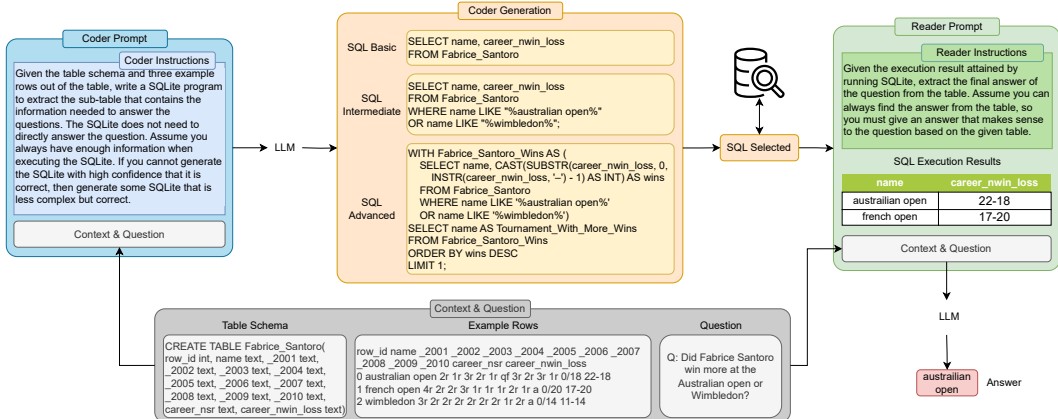

Figure 6: Concrete prompt and generation structures of both CODER and READER.

motivation. While text-to-SQL task is not the focus of this work, we carried out complementary experiments on the "flight_4" database of the Spider dataset to verify the effectiveness of our CODER module with the STC prompting strategy. Below we list the results in the order of `question`, `ground truth`, and `SQL generated by CODER`. We also show the stats of the large-scale database in the table.

```
How many routes does American Airlines operate?
SELECT count(*) FROM airlines AS T1 JOIN routes AS T2 ON T1.alid =
    ↪ T2.alid WHERE T1.name = 'American Airlines'
SELECT COUNT(*) FROM routes r INNER JOIN airlines a ON r.alid = a.
    ↪ alid WHERE a.name = 'American Airlines';

How many routes end in a Canadian airport?
SELECT count(*) FROM airports AS T1 JOIN routes AS T2 ON T1.apid =
    ↪ T2.dst_apid WHERE country = 'Canada'
SELECT COUNT(*) FROM routes WHERE dst_apid IN (SELECT apid FROM
    ↪ airports WHERE country = 'canada');

What are the names of all airports in Cuba or Argentina?
SELECT name FROM airports WHERE country  =  'Cuba' OR country  =
    ↪ 'Argentina'
SELECT name FROM airports WHERE country LIKE "%cuba%" OR country
    ↪ LIKE "%argentina%";
```

```
What are the countries of all airlines whose names start with
    ↪ Orbit?
SELECT country FROM airlines WHERE name LIKE 'Orbit%'
SELECT DISTINCT country FROM airlines WHERE name LIKE "Orbit%";

In how many cities are there airports in the country of Greenland?
SELECT count(DISTINCT city) FROM airports WHERE country = '
    ↪ Greenland'
SELECT COUNT(DISTINCT city) as airport_cities FROM airports WHERE
    ↪ country LIKE "%greenland%";
```

