# OpenReview forum: "OpenTab: Advancing Large Language Models as Open-domain Table Reasoners"
_ICLR.cc/2024/Conference — ICLR 2024 poster_

### Official Review · Reviewer_VJ1H · 2023-11-01

**Soundness:** 3 good
**Presentation:** 4 excellent
**Contribution:** 3 good
**Rating:** 6
**Confidence:** 3

**Summary:**

The paper introduces OpenTab, an open-domain table reasoning framework powered by LLMs. Unlike most LLMs which rely on knowledge stored in their parameters or retrieved from documents, OpenTab is specifically designed to utilize the rich information contained in structured data. The framework consists of several key components: a retriever to collect relevant tables, a row selector to prune only the pertinent rows of records, a Coder to generate SQL queries, and finally, a Reader to verify the answers. Experiments conducted on Open-WikiTable, WikiTableQuestion, and Feverous demonstrate that OpenTab achieves superior performance compared to previous reader and parser-based methods on tables across both open and close domains.

**Strengths:**

- The paper is clearly written and easy to follow.
- Figures 1, 3, and 4 effectively illustrate the overall pipeline and modules.
- The proposed pipeline is novel and generally reasonable for addressing the specific task at hand.
- OpenTab outperforms the previously proposed table-reasoning methods.

**Weaknesses:**

- The proposed pipeline might be applicable only to open-domain wikipedia-based tables and not to tables with a large number of rows as in Spider or BIRD. For instance, the row selector, although I agree with the motivation, can result in incorrect candidate tables as some questions may require a large number of rows. Determining the appropriate number of rows to retrieve using a heuristic seems to assume that the pipeline is not suited for dealing with complex tables. One of the main benefits of using tables is their ability to store a large number of records (mostly in close-domain scenarios). I would like to hear the authors’ thoughts on this aspect.
- Minor typos in the manuscript (e.g., by by)

**Questions:**

- In Figure 3, reader seems to appear after the SQL is selected, but in the explanation, the authors mention that the reader is used to select the SQL. Which is correct?

---

> ### Author Response · Authors · 2023-11-19
> **Response to Reviewer VJ1H**
>
> We thank the reviewer for highlighting the novelty of our proposed method, as well as the strong performance registered compared with other baseline models. We respond to reviewer’s questions and comments below.
>
> 1. Q: Row selection used, so the method cannot handle complex questions
>
> When humans are writing SQLs to query the large-scale database, one cannot memorize hundreds of thousands of row contents. Instead humans will focus on the table schema and several representative table rows as reference to write the SQL program. We argue that although only a few example rows are selected and fed into LLM for the SQL generation purpose in CODER, it can still mimic human behavior and generate solid SQLs that lead to correct question answering when the tables are large and the question is complex and based on a considerable number of rows. In our response to Reviewer 9EJC (the first reviewer, [link](https://openreview.net/forum?id=Qa0ULgosc9&noteId=nCYNTzpOo3)), we show the SQL generations of CODER and compare with human labeled SQLs. CODER only reads three selected table rows but still generate high-quality SQLs on the large-scale database of Spider dataset.
>
> 2. Q: Minor typos in the manuscript (e.g., by by)
>
> We thank the reviewer for carefully checking the draft. We updated the manuscript for correction.
>
> 3. Q: In Figure 3, reader seems to appear after the SQL is selected, but in the explanation, the authors mention that the reader is used to select the SQL. Which is correct?
>
> To clarify, READER is not responsible for the selection of SQL. READER will be input with SQL execution results, SQL program selected as well as context information.
>
> Please let us know if you have further questions or comments.

---

### Official Review · Reviewer_zsjW · 2023-11-03

**Soundness:** 4 excellent
**Presentation:** 3 good
**Contribution:** 3 good
**Rating:** 8
**Confidence:** 4

**Summary:**

This paper proposes a new method for open-domain table reasoning based on large language models. The method contains a table retriever, row selector, coder, and a reader. The table retriever is built on BM25. For the coder, the authors propose the simple-to-complex prompting strategy to generate SQL queries for three levels to resolve the complex input queries and infeasible queries by SQL itself. Then a reader module takes the context and the intermediate SQL execution results to produce the final answer to the query.

To resolve the hallucination problem of LLM when generating SQL queries, the authors also propose a generative reranking & sequential reasoning (GRSR) strategy by assessing the similarity of the generated SQL with the query so as to evade noisy tables.

The authors conduct comprehensive experiments to demonstrate the superiority of the proposed method as well as ablation studies to show the effectiveness of each proposed module.

**Strengths:**

The proposed method is novel and effective, as demonstrated by experiments. Each proposed module is well-motivated and well-ablated.

**Weaknesses:**

Some technique details are not clear.
When using BM2.5 for the retriever, are you using the full table?
For the proposed simple-to-complex strategy for Coder, how is it working with GRSR?
For the GRSR, how to deal with SQL queries grounded on multiple gold tables?

**Questions:**

Are you going to release the code?

---

> ### Author Response · Authors · 2023-11-19
> **Response to Reviewer zsjW**
>
> We thank the reviewer for acknowledging the superiority and novelty of our proposed method and the comprehensiveness of our experiments and ablation studies. Below we answer the reviewer's questions correspondingly.
>
> 1. Q: When using BM2.5 for the retriever, are you using the full table?
>
> Yes. Because BM25 is scalable to large tables even in full size, in our experiments BM25 works on full table size. Technically we linearize the table structure into text strings by join table title (if any), header, and cell strings row by row. The BM25 algorithm is built upon the rank_bm25 library [1].
>
>
> 2. Q: For the proposed simple-to-complex strategy for Coder, how is it working with GRSR?
>
> In our implementation, the STC strategy will generate three SQLs and select one for execution and input to READER. GRSR will do ranking based on the selected SQL. We have also tried to concatenate all three SQLs and did the ranking, but this approach didn’t yield better results.
>
> 3. Q: For the GRSR, how to deal with SQL queries grounded on multiple gold tables?
>
> Due to the traits of the table QA datasets, in our work we only consider a single-gold-table scenario, where grounded evidence for each question only exists in one single table instead of multiple. One proposal on this is to preprocess retrieved tables to join them into single tables before doing reasoning. We leave further extending to multiple-gold-table settings as future research direction.
>
>
> 4. Q: Are you going to release the code?
>
> Yes we will release our code upon publication of our work.
>
>
> [1] https://github.com/dorianbrown/rank_bm25
>
> Please let us know if you have further questions or comments.

---

### Official Review · Reviewer_9EJC · 2023-11-09

**Soundness:** 3 good
**Presentation:** 3 good
**Contribution:** 3 good
**Rating:** 6
**Confidence:** 4

**Summary:**

The paper proposes OpenTab, an open-domain table reasoning framework. The framework includes A) a "retriever" module to retrieve a subset of structured tables from a table corpus (using BM25), and B) a "reasoner" module that is composed of a "coder" module for generating SQL queries, a "rowselector" module to rerank rows, and a "reader" module to parse SQL execution and its accompanying context into a natural language response. The authors also propose a Generative Reranking & Sequential Reasoning (GRSR) strategy to rerank retrieved tables using SQL and query similarity, and a simple-to-complex (STC) prompting strategy for SQL generation.

The OpenTab framework contributes a method for open-domain table reasoning that has higher accuracy than existing baselines, and that is reliant only on few-shot prompting (no fine-tuning).

**Strengths:**

The paper is largely written clearly, and easy to follow. The OpenTab framework is a notable contribution to the table reasoning and semantic parsing community for its approach to open-domain reasoning. The paper's combination of several approaches into the OpenTab framework can be considered as an original application to the problem of open-domain reasoning.

**Weaknesses:**

- The paper claims to provide a new simple-to-complex prompting strategy for text-to-SQL generation using LLMs. However, the details of this prompting strategy are relegated to Appendix A.1, and it is not clear how the LLM is guided to increase the complexity of successive SQL generations. Moreover, for this claim of a new strategy to be sound, one would expect to see it applied to standard text-to-SQL benchmarks like Spider or the new BIRD-SQL.

- The paper's framework for retrieval and generation for open-domain table reasoning is analogous to retrieval-augmented LLMs, but there is no mention of RAG (for e.g. [Lewis et al., 2020](https://arxiv.org/abs/2005.11401)). The paper is thus strangely not situated within the RAG literature (but the paper contains fair contributions to RAG, such as GRSR).

**Questions:**

**Questions**:

1. The ICL prompt for SQL generation includes this line: "Generate 3 SQLite programs with respect to the question separated by [SQLSEP], the complexities of the SQLite programs generated ascend (basic, intermediate, advanced)." to "sequentially generate three SQL programs with ascending complexities", SQL-basic, SQL-intermediate and SQL-advanced.

    - a) This prompt seems intuitively underspecified in guiding the model to generate SQL that focuses on column selection in the first SQL, and column+row selection in the second SQL. In other words, the wording of the prompt doesn't specify what kind of SQL should be generated in each stage. How did the authors verify the SQL was actually of that specific functionality?

    - b) How did the authors verify that the SQL written was actually of ascending complexity?

2. SQL generation performance: In claiming the new simple-to-complex prompting strategy for SQL generation, how did the authors verify the performance of this method on text-to-sql benchmarks like Spider or BIRD-SQL?

3. Lost in the middle phenomenon: Was there any prompt engineering done for the ICL prompts to counter the "lost in the middle" phenomenon, particularly for the reader module which incorporates substantial context?

4. From Section 4.2, given that the baseline methods "require full tables to be fed into LLM for reasoning, which may exceed the upper bound of the input token limit, rendering an invalid prediction.", do the results for baseline methods include or exclude these invalid predictions?

5. From Table 4, what explains the large gap in difference for BM25 and BM25* for the "w/o Fine-tuning" row? This seems surprising given that BM25 is just supposed to be the authors' replication of BM25* presented in Kweon et al., 2023.

6. On the effectiveness of GRSR: how does the effectiveness of GRSR vary with different choices of top-k (where k is number of tables retreived)?


**Suggestions**:
- In Section 3.1, "DPR" is first used. Suggest to include the full name of Dense Passage Retrieval when it is first used, and use the acronym subsequently.
- In Section 5, explain why using all three SQL difficulties leads to the best performance, as was done in Section 3.2.1
- It is likely relevant for the authors to include a section on retrieval augmented generation in LLMs under Section 6.

---

> ### Author Response · Authors · 2023-11-19
> **Response to Reviewer 9EJC #1**
>
> We appreciate the reviewer for acknowledging the notable contributions by our work and the novelty of OpenTab to the important Open-domain table reasoning task. Below we address the reviewer's concerns accordingly.
>
>
> 1. Q: The paper claims to provide a new simple-to-complex prompting strategy for text-to-SQL generation using LLMs, …and it is not clear how the LLM is guided to increase the complexity of successive SQL generations
>
> We guided LLM to generate three increasingly complex SQL programs by two means.
>
> - Firstly in the prompt we have such instruction, “Generate 3 SQLite programs with respect to the question separated by [SQLSEP], the complexities of the SQLite programs generated ascend (basic, intermediate, advanced).”. Such instruction works effectively on the instruction-tuned LLMs we studied, which are gpt-3.5-turbo and falcon-180b.
> - What’s more we included few-shot examples in the prompt, showing that generated SQLs are in three complexities reflecting different functionalities in order. LLMs learned from the demonstrations well because of their in-context learning ability.
>
>
> 2. Q: …prompt seems intuitively underspecified in guiding the model to generate SQL that focuses on column selection in the first SQL, and column+row selection in the second SQL. In other words, the wording of the prompt doesn't specify what kind of SQL should be generated in each stage. How did the authors verify the SQL was actually of that specific functionality?
>
> As we have explained in the last question, the SQL generation is instructed and guided by the prompt instruction as well as the few-shot demonstrations. Although we didn’t explicitly map [simple, intermediate, complex] to specific functionalities in the instruction, such correlation can be implicitly recovered in the given few-shot demonstrations. Moreover, we designed such demonstrations with the help of gpt-4. We asked gpt-4 to generate SQLs following the [simple, intermediate, complex] types and got back the initial SQL drafts reflecting the different functionalities (row selection, column selection, and beyond), intuitively showing that there could be some implicit alignment between the keywords and the functionalities.
>
>
> 3. Q: Moreover, for this claim of a new strategy (STC) to be sound, one would expect to see it applied to standard text-to-SQL benchmarks like Spider or the new BIRD-SQL.
>
> STC is proposed in the context of end-to-end table QA task, in which SQL programs work as an efficient and scalable interface to query the database and extract relevant information for READER to make inference on. SQL generation is an intermediate tool in the QA pipeline, instead of outcome. While on text-to-SQL benchmarks, ground truth is SQL program itself, which is different from our motivation. While text-to-SQL task is not the focus of this work, we carried out complementary experiments on the “flight_4” database of the Spider dataset to verify the effectiveness of our CODER module with the STC prompting strategy. Below we list the results in the order of question, ground truth, and SQL generated by CODER. We also show the stats of the large-scale database in the table.
>
>
> | Table name | Row number |
> | -------- | -------- |
> | airlines | 6162     |
> | airports | 7184 |
> | routes | 67240 |
>
>
> ```
> How many routes does American Airlines operate?
> SELECT count(*) FROM airlines AS T1 JOIN routes AS T2 ON T1.alid = T2.alid WHERE T1.name = 'American Airlines'
> SELECT COUNT(*) FROM routes r INNER JOIN airlines a ON r.alid = a.alid WHERE a.name = 'American Airlines';
> ```
>
> ```
> How many routes end in a Canadian airport?
> SELECT count(*) FROM airports AS T1 JOIN routes AS T2 ON T1.apid = T2.dst_apid WHERE country = 'Canada'
> SELECT COUNT(*) FROM routes WHERE dst_apid IN (SELECT apid FROM airports WHERE country = 'canada');
> ```
>
>
> ```
> What are the names of all airports in Cuba or Argentina?
> SELECT name FROM airports WHERE country  =  'Cuba' OR country  =  'Argentina'
> SELECT name FROM airports WHERE country LIKE "%cuba%" OR country LIKE "%argentina%";
> ```
>
> ```
> What are the countries of all airlines whose names start with Orbit?
> SELECT country FROM airlines WHERE name LIKE 'Orbit%'
> SELECT DISTINCT country FROM airlines WHERE name LIKE "Orbit%";
> ```
>
> ```
> In how many cities are there airports in the country of Greenland?
> SELECT count(DISTINCT city) FROM airports WHERE country = 'Greenland'
> SELECT COUNT(DISTINCT city) as airport_cities FROM airports WHERE country LIKE "%greenland%";
> ```
>
> We see that our CODER is able to generate solid SQL programs that achieve equivalent functionalities as human labeled, demonstrating the efficacy of our CODER module.

---

> ### Author Response · Authors · 2023-11-19
> **Response to Reviewer 9EJC #2**
>
> 4. Q: Didn’t do literature review on RAG papers
>
> Thanks for the suggestion. We drafted a paragraph of discussion about the related works on RAG in Section 6 of the draft. We also put it here for review.
>
> Retrieval Augmented Generators. RAGs [6] leverage retrievers to fetch information from external knowledge database and augment the text input to the language models. REALM [5] and RETRO [1] pretrains the retrieval augmentation framework in the end-to-end manner so that the language model learns to better utilize the retrieved information, while ATLAS [2] jointly trains the retriever as well as the language model. We point out that these well-known RAG models are specialized in the text reasoning domain and cannot be directly applied to table reasoning tasks without fine-tuning, thus inapplicable to the experimental setup of this work.
>
>
>
> 5. Q: Lost in the middle issue considered?
>
> As in Liu et al. 2023 [7], the “lost in the middle” problem is based on the severe long-context issue of LLM. For OpenTab in the open-domain setting, where the framework needs to inference on multiple retrieved tables, the JR (joint reasoning) strategy has struggling accuracy compared with SR (sequential reasoning) and GRSR (generative reranking & sequential reasoning). Considering the long-context trait of JR, the “lost in the middle” issue could be the cause that is impacting the performance of JR. While for the default strategy GRSR, each time OpenTab does reasoning over one retrieved table and later collectively carries out generative reranking, so that we can effectively evade the “long context” issue and “lost in the middle” problem.
>
>
> 6. Q: From Section 4.2, given that the baseline methods "require full tables to be fed into LLM for reasoning, which may exceed the upper bound of the input token limit, rendering an invalid prediction.", do the results for baseline methods include or exclude these invalid predictions?
>
> For the baseline methods, we include the invalid predictions and treat them as wrong predictions as scalability to large tables is an important property for advanced table reasoners. In experiment most tables in Open-WikiTables and FEVEROUS won’t exceed the 4k token limit of gpt-3.5-turbo and falcon-180b with 2-shot demonstrations. The most severe token overflow issue happens on BINDER, which by default uses 14-shot examples. To fit the long sequence we deploy gpt-3.5-turbo-16k as backbone to run the BINDER.
>
>
> 7. Q: From Table 4, what explains the large gap in difference for BM25 and BM25* for the "w/o Fine-tuning" row? This seems surprising given that BM25 is just supposed to be the authors' replication of BM25* presented in Kweon et al., 2023.
>
> As in the official repo of Kweon et al, 2023 [3,4], there is no public implementation of the BM25 experiments. In the paper the authors did not disclose details of their implementation, either. Due to these reasons we cannot fully reproduce their results. We implemented BM25 on our side following the setting of Kweon et al, 2023 to the most extent and reported the numbers in Table 4. We will open source our code when our work gets published.
>
>
> 8. Q: On the effectiveness of GRSR: how does the effectiveness of GRSR vary with different choices of top-k (where k is number of tables retrieved)?
>
> We refer the reviewer to the third paragraph of Section 5, where we did ablation study to verify the effectiveness of GRSR with varying k. In Figure 5, GRSR improves the QA accuracy with more tables retrieved, while the plain JR and SR methods got more distracted and might get worse accuracy when k increases.
>
>
> 9. Q: In Section 5, explain why using all three SQL difficulties leads to the best performance, as was done in Section 3.2.1
>
> For our STC strategy, SQLs are incrementally generated, and the generation path can also function as reasoning steps, making the generations more likely valid. Moreover the solidness of the generated SQL programs will be verified on the fly, thus we can try to evade invalid SQLs that will either have syntax errors or empty execution results, which is common for a solo SQL generation. To sum up, the STC prompting strategy improves the performance through incremental reasoning as well as flexibility.

---

> ### Author Response · Authors · 2023-11-19
> **Response to Reviewer 9EJC #3**
>
> [1] Borgeaud, Sebastian, et al. "Improving language models by retrieving from trillions of tokens." International conference on machine learning. PMLR, 2022.
>
> [2] Izacard, Gautier, et al. "Few-shot learning with retrieval augmented language models." arXiv preprint arXiv:2208.03299 (2022).
>
> [3] Kweon, Sunjun, et al. "Open-WikiTable: Dataset for Open Domain Question Answering with Complex Reasoning over Table." arXiv preprint arXiv:2305.07288 (2023).
>
> [4] https://github.com/sean0042/Open_WikiTable
>
> [5] Guu, Kelvin, et al. "Retrieval augmented language model pre-training." International conference on machine learning. PMLR, 2020.
>
> [6] Lewis, Patrick, et al. "Retrieval-augmented generation for knowledge-intensive nlp tasks." Advances in Neural Information Processing Systems 33 (2020): 9459-9474.
>
> [7] Liu, Nelson F., et al. "Lost in the middle: How language models use long contexts." arXiv preprint arXiv:2307.03172 (2023).
>
> Please let us know if you have further questions or comments.

---

### Author Response · Authors · 2023-11-19
**Draft updated**

Dear reviewers,

We updated our draft with the following modifications

- Corrected several typos for better readability.
- Included a paragraph of related work discussion on RAGs.
- Added full name of DPR method.

Thank you.

---

### Meta-Review · Area_Chair_95JX · 2023-12-10

**Metareview:**

This paper presents a method to perform reasoning over tables using LLMs. The overall task is broken into intermediate steps of (1) table retrieval, (2) selection of relevant information from the table, and (3) generation of response. There are methodical improvements present in #2 that lead to SOTA results on existing benchmarks for table QA. The paper is well-written and the experiments are through and sound. For open domain QA, where the table to be QAed from isn't pgiven in advanced, there is a risk of incorrect tables being retrieved and for that the authors present a method using an existing cross encoder to rank the generated SQL against the table. This helps in removing the tables that aren't relevant to the query. The paper contains improvements to a bunch of different modules in the entire QA pipeline that leads to the overall improvement of results.

**Justification For Why Not Higher Score:**

The paper presents some obvious to do improvements in the pipeline. There are some open questions on the ability of the model because the model extracts rows instead of reading entire table, and the evaluation benchmarks can be extended.

**Justification For Why Not Lower Score:**

Clearly written paper with some novelty and improvement of SOTA on some benchmarks.

---

### Decision · Program_Chairs · 2024-01-16

Accept (poster)